# Botulinum Toxin for the Treatment of Raynaud’s Conditions of the Hand: Clinical Practice Updates and Future Directions

**DOI:** 10.3390/toxins16110472

**Published:** 2024-11-01

**Authors:** Patrick O’Donohoe, Jake McDonnell, Justin Wormald, Lylas Aljohmani, Kevin Cronin, Laura Durcan, Oran Kennedy, Roisin Dolan

**Affiliations:** 1Department of Plastic & Reconstructive Surgery, St. Vincent’s University Hospital, D04T6F4 Dublin, Ireland; 2Department of Surgery, Royal College of Surgeons in Ireland, D02YN77 Dublin, Ireland; 3Nuffield Department of Orthopaedics, Rheumatology and Musculoskeletal Sciences, University of Oxford, Oxford OX1 2DJ, UK; 4Department of Plastic & Reconstructive Surgery, John Radcliffe Hospital, Oxford OX3 9DU, UK; 5Department of Plastic & Reconstructive Surgery, Mater Misericordiae University Hospital, D07R2WY Dublin, Ireland; 6Department of Rheumatology, Beaumont Hospital, D09V2N0 Dublin, Ireland; 7Tissue Engineering Research Group, Department of Anatomy & Regenerative Medicine, Royal College of Surgeons in Ireland, D02YN77 Dublin, Ireland; 8Trinity Center for Biomedical Engineering, Trinity College Dublin, D02PN40 Dublin, Ireland; 9Advanced Materials and Bioengineering Research Centre (AMBER), D02PN40 Dublin, Ireland; 10UCD School of Medicine & Medical Sciences, University College Dublin, D04V1W8 Dublin, Ireland

**Keywords:** botulinum toxin, Raynaud’s phenomenon, Raynaud’s disease, patient-reported outcomes

## Abstract

Raynaud’s conditions of the hand, referred to commonly as Raynaud’s phenomenon, both primary and secondary, represents a spectrum of disorders affecting the digits, characterised by recurrent episodes of vasospasm that result in a triad of symptoms: pain, pallor, and cyanosis. Various therapies, ranging from conservative hand therapy techniques to surgical sympathectomy, have been explored with inconsistent results. Recently, the local administration of botulinum toxin type-A (BTX-A) has re-emerged as a treatment option for this condition. This review delves into the mechanistic pathways of BTX-A therapy, optimal dosing concentrations, administration techniques, and its safety profile. A critical analysis of published studies to date demonstrates varied clinical efficacy of BTX-A in Raynaud’s conditions based on patient-reported outcome measures and objective measures of outcomes assessment. Thus, in order to accurately assess the clinical effectiveness of BTX-A in future robust studies, this review emphasises the importance of streamlining patient selection to minimise heterogeneity in disease severity, optimising recruitment to ensure adequate statistical power, and establishing sensitive outcome measures to monitor response and discern treatment efficacy. Additionally, addressing concerns such as minimising antibody resistance, extending the duration of treatment effects on tissues, and exploring new modalities to assess hand perfusion will be focal points for future research and BTX-A drug development.

## 1. Introduction

Raynaud’s conditions of the hand, with and without an associated underlying autoimmune disease, were first described by Maurice Raynaud in 1862 [1]. He described recurrent episodic vasospasm affecting the digits, presenting with a triad of pallor, cyanosis, and pain [1,2,3].

This is a common phenomenon. The estimated overall prevalence in the general population is 3–5%, with a higher incidence reported in female patients, with associated emotional stressors, and in colder climates [3]. Primary Raynaud’s phenomenon (PRP) describes triphasic colour change in the absence of an associated autoimmune disease and is more common than secondary (SRP), although changes in classification can occur with evolving connective tissue disorders [3]. While the rates of PRP vs. SRP are underreported, a meta-analysis of *n* = 639 patients initially diagnosed with PRP revealed that 12.6% were subsequently re-classified as SRP during clinical surveillance [4].

Patients with PRP typically experience less pain, have normal nailfold capillaries reflecting a lack of vascular damage, and lack an associated condition such as rheumatoid arthritis, systemic sclerosis, or systemic lupus erythematosus [1]. Patients demonstrating the presence of Raynaud’s with positive autoantibodies, without a frank diagnosis of a connective tissue disorder, are considered at higher risk for evolution to a secondary diagnosis than those without evidence of autoimmunity on laboratory testing.

Patients with Raynaud’s associated with a connective tissue disease are at a higher risk of significant disease-related complications, reflecting underlying vasculopathy. These include digital pitting, ulceration, ischaemia and gangrene, resulting in impaired hand function and negatively impacting their ability to work and look after themselves [3,5].

The pathology of Raynaud’s phenomenon differs slightly between the two disease entities. In PRP, the exaggeration of physiological vasospasm overrides vasodilatation in the digits [3]. While also present in SRP, there is often associated vascular dysfunction, as evidenced by vascular occlusion from microthrombi, endothelial dysfunction, and platelet activation [6].

Current treatments for RP include lifestyle modification with an avoidance of triggers [7] and pharmacological interventions such as calcium channel blockers (CCBs) [8], PDE-5 inhibitors, topical nitrates, ACE inhibitors, angiotensin receptor blockers (ARBs), selective serotonin reuptake inhibitors (SSRIs), endothelin-1 antagonists, and prostacyclin [1,3,7]. These have variable efficacy and are often poorly tolerated.

Treatment protocols targeting Raynaud’s, both primary and secondary, typically commence with lifestyle modifications and monotherapy with CCBs, replaced by ARBs, SSRIs, or PDE-5 inhibitors if needed. Patients with severe symptoms or any ischaemic ulceration often progress to combination therapy with CCBs, PDE-5 inhibitors, and aspirin. Those experiencing recurrent severe events and ischaemic ulceration may be considered for endothelin-1 antagonists and prostacyclin [1,3,5]. Surgical interventions are often explored in severe Raynaud’s, which has been resistant to conservative and pharmacological therapies. These interventions, such as sympathetic trunk transection and digital sympathectomy, are not without complications, including compensatory hyperhidrosis and disease recurrence [9]. To address the significant therapeutic gap that exists for these patients, there is a growing role for the local delivery of botulinum toxin type-A (BTX-A) to treat this challenging condition.

Botulinum neurotoxin (BTX) is a product of the Gram-positive bacillus Clostridum Botulinum [6]. Subcutaneous injections of BTX-A have been sporadically trialled in the management of treatment-refractory Raynaud’s conditions of the hand for nearly 20 years [10]. BTX-A, a commercially available form of the neurotoxin, is employed for its effect in blocking the release of acetylcholine from peripheral cholinergic nerve terminals, offering an option for the treatment of disorders characterised by excessive cholinergic neuronal activity [11]. Recently, BTX-A has resurfaced as a potential treatment strategy for this challenging condition [1,12,13,14].

In this narrative review, we delve into the mechanistic therapeutic pathways of BTX-A, minimum effective dosing concentrations, optimal administration techniques, and the pharmacological safety profile of BTX-A. We discuss various published study designs investigating the clinical effectiveness of the subcutaneous administration of BTX-A, highlighting the inconsistent reporting of the clinical efficacy of BTX-A using both patient-reported outcome measures (PROMs) and Objective measures of outcomes assessment (OMOA) such as perfusion studies, thermography, and pulse pressures. Finally, we suggest potential avenues of interest for future BTX-A research and drug development, which have the potential to maximise its therapeutic benefit.

### 1.1. Botulinum Toxin: Proposed Mechanism of Action

The BTX-A products most commonly utilised in clinical practice are onabotulinumtoxinA (marketed as Botox^®^) and abobotulinumtoxinA (marketed as Dysport^®^), although other preparations exist, including incobotulinumtoxinA (marketed as Xeomin^®^) and the BTX-B product rimabotulinumtoxinB (marketed as Myobloc^®^). Only BTX-A and -B are approved for human use—BTX-A targets SNAP-25 while BTX-B targets VAMP [11,15,16]. These agents have been employed as therapeutic agents in various conditions, including autonomic disorders, spasticity, migraine and neuropathic pain, the cosmetic treatment of rhytides, hyperhidrosis, overactive bladder, and Raynaud’s conditions [6,11,15,16,17,18].

Various theories propose the mechanism of clinical action of BTX-A in Raynaud’s conditions. At a fundamental level, this neurotoxin targets various molecular components within the presynaptic neuron, including the soluble NSF attachment protein receptor (SNARE) complex, vesicle-associated membrane protein (VAMP), syntaxin, and synaptosomal-associated protein (SNAP-25) [11] (see Figure 1). It is suggested that the BTX-A-related inhibition of sympathetic nerve action reduces vascular smooth muscle contraction, thereby enhancing blood flow to the digit [7,10,16,19,20,21]. With this mechanism of action in mind, botulinum toxin has been described as an alternative to sympathectomy, and we note that Vegas et al. reported a case combining sympathectomy and botulinum toxin administration [22].

Studies also postulate that BTX-A may influence local physiology to promote wound healing by decreasing inflammatory cell infiltration [23,24], enhancing blood flow [20,25,26], reducing noradrenaline release (hence increasing vasodilation and oxygen delivery to the wound) [24], decreasing the generation of reactive oxygen species and endothelial nitric oxide synthase in the wound bed [27], and reducing the expression of TGF-β1 [23,28]. Other theories have also suggested BTX-A playing a role in the prevention of exocytosis of endothelin-1 and von Willebrand Factor, which may have a role in reducing microthrombosis in capillaries and subsequent ischemia and pathological remodelling of digital blood vessels [1,6,7,17,29].

Additionally, the inhibition of Substance P and Calcitonin gene-related peptide (CGRP) release by sensory fibres is proposed to contribute to symptom relief [7,15,16,17,29]. Both Substance P and CGRP are neuropeptides that are known to affect immune responses in inflammatory conditions such as psoriasis or Raynaud’s phenomenon [7,15,17,29,30]. Popescu et al. discuss in depth in their review paper how botulinum toxin may be effective in treating psoriasis through the modulation of these neuropeptides [30]—it is possible that this is the origin of botulinum toxin’s efficacy in Raynaud’s phenomenon.

Twenty years ago, Sycha et al. [10] were the first to describe the use of BTX-A treatment in *n* = 2 patients with treatment-resistant Raynaud’s conditions (one SRP and one PRP). The subjective outcome assessment included pain, stiffness, and numbness (Visual Analogue Scale). Objective measures included superficial skin blood flow at 6 weeks post-injection as measured by laser Doppler imaging. Both patients demonstrated significant improvement in outcome measures at follow-up [10].

### 1.2. Botulinum Toxin: Administration Techniques

Various techniques have been employed in the treatment of Raynaud’s conditions with BTX-A [13] (Table 1). Gallegos et al. outlined their preferred approach involving reconstitution of a 100-unit vial of onabotulinumtoxinA (Botox^®^) with 20 mL of normal saline, resulting in a concentration of 5 IU/mL. This solution is then divided into 2 mL injections, with each hand receiving five injections, totalling 10 IU per finger and 50 IU per hand [17]. A similar technique is applied to a 300-unit vial of abobotulinumtoxinA (Dysport^®^), providing dosages of 30 IU per finger and 150 IU per hand [17]. Dosing regiments vary, with Neumeister et al. recommending 10 IU of Botox^®^ per digit [14,19], while Nagarajan et al. described variable doses depending on the severity of the patient’s condition [31].

Evidence relating to the dose/response relationship of BTX-A is limited. Iorio et al.’s 2012 review [21] emphasised the lack of studies comparing dose to reported patient outcomes, and a 2021 review by Ennis et al. highlighted the overall inadequacy of evidence due to under-powering and poor study design [6]. The most commonly studied maximum dose is 100 IU of BTX-A per hand [6].

Injection sites are commonly found in the distal palm and digits, with a palmar approach at the A1 pulley [13,14,16,17,19,32] or metacarpophalangeal joint (MCPJ) [6,13,29] being frequent. Some studies, such as the paper by Lobb et al., propose ultrasound guidance to enhance precision, although evidence is lacking regarding the superior clinical efficacy of this approach [33]. Dorsal injection techniques [6,12,13] and injections at the volar wrist level [31,34] have been explored, but their benefits and potential adverse effects are not clearly defined [6,13,35]. Fregene et al. found no statistically significant difference in outcomes comparing interdigital space injections to distal palmar injections, although the data favoured distal palm injections [36].

The ideal injection frequency lacks consensus, but Gallegos et al. suggest injections at 1, 3, and 6 months [17]. While some trials have assessed patient outcomes based on a single administration of BTX-A [12,13,16,29,37], its temporary effect, lasting about 2–4 months [11], should be considered when evaluating outcomes. The onset of action occurs 1–4 days post-injection [11,19], with common transient side effects, including intrinsic muscle weakness and injection site pain [6,13,19].

### 1.3. Botulinum Toxin: Clinical Efficacy

The current body of literature on the effectiveness of BTX-A in Raynaud’s conditions of the hand reveals a deficiency in high-quality, adequately powered, prospective studies with consistent outcome measures. The recent TORCH study by Geary et al. [38] aimed to address this gap by synthesising and critically appraising the existing literature. Among the 18 studies included, BTX-A treatment exhibited a significant association with an overall mean improvement in pain Visual Analogue Scale scores (4.11+/− SD: 2.4) and a mean percentage of digital ulcer healing (88.1%, SD: 17.6). The meta-analysis indicated a high probability of pain score improvement (81.95% (95% CI [74.12–87.81]) and digital ulcer healing (79.37%, 95% CI [62.45–89.90]). However, retrospective designs, inadequate power, and heterogeneity in study cohort characteristics limited the robustness of these findings [12,19,31,32,34,36,37,39,40,41,42,43,44,45,46,47,48,49]. Notably, variations in injection sites and BTX-A dose concentrations across studies were evident, as highlighted by Lawson et al. and Iorio et al. [13,21,38] (Table 1).
toxins-16-00472-t001_Table 1Table 1Characteristics of *n* = 18 included studies in the TORCH study [38].AuthorYearMethods NInjection Site/Target (Note Volar Approach Unless Otherwise Stated)BTX Dose per Hand (U)Mean Follow-Up (Months)(Mean)Outcomes MeasuredFindings Van Beek et al. [46] 2007Retrospective case series11Mixed-Proper Digital Artery, Common Digital Artery, and Superficial Palmar arch all used50–2009.6Healing of Digital UlcersVAS scoreIt was found that 81% of patients had complete resolution of pain and healing of digital ulcers, and 100% had improvement in VAS scores.Fregene et al. [36] 2009Retrospective case series26Mixed-Proper Digital Arter, Superficial Palmar Arch, and Volar Wrist all used20–10018Pain rating, Colour and Appearance of Digits, Transcutaneous Oxygen Saturation, Ulcer HealingIt was found that 75% of patients experienced a change in their VAS score, and 47% reported healing of digital ulcers.Neumeister et al. [19]2010Retrospective case series33Neurovascular Bundle at level of A1 Pulley37(22–57)10–72Pain, Healing of Digital UlcersIt was found that 85% experienced improvement in VAS score, and 100% reported healing of digital ulcers.Jenkins et al. [42]2013Randomised control trial8Superficial Palmar Arch Proximal to A1 Pulley40-Digital pulp temperatureImprovement in digital temperature.Serri et al. [45] 2013Prospective case series18Proper Digital Arteries Distal to A1 Pulley100-Healing of Digital Ulcers, QuickDASH Score, Oxygen partial pressure, painIt was found that 100% reported healing of digital ulcers at one month.Uppal et al. [37]2014Prospective non-randomised control trial20Digital Neurovascular Bundles at level of distal Palmar Crease1006Pain, appearance, cold intolerance, pinch and power grip, ranges of movement, DASH scoreIt was found that 75% reported healing of digital ulcers, and 80% reported improvement in VAS scores. Zhang et al. [48]2015Retrospective case series10Neurovascular Bundles at level of MCPJs506Surface temperature, VAS score, digital ulcer healingIt was found that 100% of patients experienced improvement in VAS score and healing of digital ulcers.Motegi et al. [32]2018Prospective case series10Neurovascular Bundles just proximal to A1 Pulley104VAS, temperatureIt was found that 75% reported improvement in VAS, and 100% reported healing of digital ulcers. Bello et al. [12]2017Randomised control trial40Common Digital Arteries to each finger, Dorsal Approach504Digital arterial velocity flow, QuickDASH, McCabe Cold Sensitivity Score, VAS, Raynaud’s Condition ScoreAll metrics showed some improvement; none reached statistical significance.Medina et al. [43]2018Retrospective case series15Lateral aspects of the base of each digitMean 43.636VAS, digital ulcer healingAt 3–6 months, 36% maintained a decreased VAS score, and 71% reported healing of digital ulcers. Dhaliwal et al. [40] 2018Prospective case series40Toes rather than fingers-injections around neurovascular bundles503Ulcer Healing, Foot temperature and swelling, frequency of attacksAt 6 weeks post-injection, all patients reported improvement in pain and cold intolerance.Aarthi et al. [39] 2020Randomised control trial2010 sites per hand, not detailed where503VAS score, DASH, Digital Ulcer HealingMean VAS in the treatment group was significantly reduced at 3 months. Habib et al. [41]2020Case series3Injections into web spaces of each digit at level of MCPJs dorsally320.5Pain score, symptomatic improvement, digital ulcer healingImprovement in pain and cold tolerance was found in 2/3 of patients.Winter et al. [47]2020Case series4Mixed-Distal Palmar Crease and Dorsal Webspace of each digit both injected10–10016.25VAS scoreDecreased pain in each patient was found to varying degrees.Nagarajan, et al. [31] 2021Retrospective case series11Mixed-Digital Neurovascular Bundles in Palm, Radial/Ulnar Arteries at wrist all used (for foot patients Plantar NVB or Doralis Pedis/Posterior Tibial Artery at ankle)20–50049Symptom relief, digital ulcer healingIt was found that 100% of patients reported improvement in pain and healing of digital ulcers. Goldberg et al. [34]2021Retrospective case series20Mixed-Injections performed both at levels of fingers and wrist10010.5VAS, frequency of episodes, quick DASHIt was found that 88% reported reduction in VAS and mean reduction in quickDASH by 22.4 at 6 months.Shenavandeh et al. [49]2022Non-randomised control trial26Proper digital arteries of all digits1001VAS and RP, skin colour and type of ulcers, capillaroscopy It was foun that 95.5% reported improvement in digital ulcer healing.Senet P, et al. [44]2023Randomised control trial90Neurovascular bundles to each digit at level of distal palm505Raynaud’s Condition Score, Health Assessment Questionnaire disability index, QuickDASH, Cochin Hand Function ScaleImprovement in symptoms was found but was not statistically significant compared to placebo.


Although all 18 studies reported clinical improvements in symptoms following BTX-A treatment, the lack of standardised measures and inconsistent reporting impede the development of evidence-based consensus guidelines for its use in Raynaud’s conditions of the hand. We note that the current research in this area is often underpowered, and, indeed, the main recommendation by Geary et al. was for larger high-powered randomised controlled trials with better reporting of outcomes and complications [38].

In the interim, BTX-A should be considered as part of the treatment armamentarium for Raynaud’s disease, particularly in cases of refractory painful acute digital ulceration, in combination with meticulous surgical debridement of necrotic tissue. There is an urgent need for pre-RCT (randomised controlled trial) feasibility studies to inform injection protocols, establish minimum effective BTX-A dose concentrations, achieve adequate power for sub-cohorts with a poorer prognosis (SSc-RP, scleroderma-associated secondary Raynaud’s), and identify reliable and sensitive outcome measures. These efforts are crucial for the accurate design of an RCT that will accurately assess the efficacy of BTX-A in Raynaud’s disease of the hand.

Safety concerns have also been noted, with complications often underreported [6,13]. Lawon et al. estimated minor complications in 20.2% of patients, predominantly in those with systemic sclerosis, mostly involving transient pain or weakness at higher risk [13]. Serious complications are rare, with no clear association between the injection site/method and complication rate described in the literature [13]. Ennis et al. reported intrinsic muscle weakness as the most common complication in their review of 33 studies, affecting 32 of 421 patients (7.6%), with injection site pain noted in 2 patients out of 421 [6]. Injection site pain should not be discounted given that the location of these injections, such as the palm and fingers, are often highly sensitive, and patients should be counselled on discomfort during each injection. Overall, however, the side effect profile compares favourably to that of systemic pharmacological agents, which can induce substantial cardiovascular side effects [1,3,6,7].

Furthermore, clinical resistance and the loss of treatment effectiveness with BTX-A are rare phenomena characterised by sub-optimal responses, termed primary or secondary non-responsiveness. This is believed to stem from the development of antibodies that neutralise BTX-A, although the exact relationship between antibody development and resistance remains unclear [50]. Although this phenomenon has not been extensively studied in the context of Raynaud’s conditions, the presence of antibodies has been attributed to shorter dosing intervals, higher doses per injection cycle, and higher amounts of antigenic protein. Some newer formulations featuring purified core neurotoxin without accessory proteins may exhibit lower overall immunogenicity [50]. Addressing antibody-associated non-responsiveness poses challenges in prevention and treatment, involving the use of less immunogenic BTX formulations, waiting for the spontaneous disappearance of the neutralising antibody, and switching to an alternate type of BTX.

Current approaches to manage this phenomenon include increasing the length of time between injections [51,52], developing less immunogenic preparations of botulinum toxin such as incobotulinumtoxinA [53], and the development of cost-effective assays to measure antibody levels earlier so that treatment can be modified to prevent the development of Secondary Non-Response [50,53,54]. Ongoing research in this area focuses on laboratory assessments of protein structure and function in addition to clinical studies comparing dosing and treatment intervals to the development of non-response. Clinicians planning on utilising BTX in the treatment of Raynaud’s phenomenon should stay abreast of developments in this area.

### 1.4. Clinical Outcomes Assessment: Challenges and New Approaches

A diverse array of outcome measures has been proposed to assess the severity of symptoms in Raynaud’s conditions and to quantify and compare treatment response. These measures fall into two categories: patient-reported outcome measures (PROMs) and objective assessments of digital perfusion (OADP). While individual measures have demonstrated clinical utility, the multitude of reported measures presents a challenge for those endeavouring to critically synthesise results across multiple studies and cohorts. A breakdown of these measures is shown below in Figure 2.

#### 1.4.1. Patient-Reported Outcome Measures (PROMS)

PROMs have gained recent prominence as subjective primary outcomes in studying hand conditions (symptoms, health-related quality of life, and functional status) in order to accurately capture the impact of a disease or treatment on a patient [55]. In the context of Raynaud’s conditions, effective outcome measurement is critical to quality of care and to conduct clinically meaningful research. Several PROMs have been shown to be useful in the assessment of Raynaud’s conditions. These include disease-specific PROMs, i.e., scoring systems specific to Raynaud’s for addressing pain, ulceration, and function, such as Raynaud’s Condition Score [12,44] and the Assessment of Systemic Sclerosis-Associated Raynaud’s Phenomenon (ASRAP) [56]), both of which have undergone validation and are considered reliable tools [56,57]. Additionally, the patient’s self-recorded frequency and intensity of vasospasm episodes are considered.

More generic PROMs include functional assessment scores such as Quick Disabilities of the Arm, Shoulder and Hand (QuickDASH) or the formal DASH score [12,34,37,44,45], the Cochin Hand Function Scale [44], the Jebsen Hand Function Test (JHFT) and the Michigan Hand Outcomes Questionnaire (MHOQ) [58]. These tools are not specific to Raynaud’s conditions but have been used extensively in the research of conditions affecting hand function, and their sensitivity and accuracy have been extensively discussed previously [58,59].

Generalised health-related quality of life PROMs such as the Patient Acceptable Symptom State (PASS) [6] and Health Assessment Questionnaire Disability Index [44] have been utilised in outcomes assessment following treatment for Raynaud’s disease. We note that although these are validated assessment tools, as the assessment becomes less specific compared to some of the previously discussed PROMS, these tools may struggle to capture small differences or improvements between treatment modalities. Pain-specific PROMs previously described in Raynaud’s include the Visual Analogue Scale (VAS) [12,32,34,39,43,46,47,48,49] and the McCabe Cold Sensitivity Score [12]. Both scales have been validated clinically and are used extensively in research on the upper limb [60,61].

Accurately capturing PROMs is an essential step in quantifying a minimal important difference (MID) to power clinical trials appropriately. Researchers assessing the clinical effectiveness of BTX-A in Raynaud’s conditions should involve patient focus groups to analyse existing PROMs or develop new ones that accurately reflect patients’ lived experiences, priorities, and preferred terminology.

#### 1.4.2. Objective Measures of Digital Perfusion

The evaluation of digital perfusion in Raynaud’s conditions has utilised diverse objective measurement techniques, including nailfold capillaroscopy, laser Doppler imaging (LDI), high-resolution Doppler ultrasound, thermographic imaging, indocyanine green (ICG) fluoroscopy, CT angiography, transcutaneous oxygen saturation/partial pressure, clinical assessments of digital ulcer healing, and skin temperature measurement [36,45,62,63,64,65,66,67,68]. Anticipating the mechanisms of BTXA-induced reduction in vascular smooth muscle contraction and noradrenaline release, there is a clinical expectation of an increase in digital perfusion.

Nailfold capillaroscopy, employing video capillaroscopy, USB microscopy, or a dermatoscope, is a frequently used perfusion assessment technique in Raynaud’s conditions and is often quantified using the Reynolds Score (RS) [64,65,67,69,70]. Capillaroscopic changes indicating microvascular damage include widened capillaries, microhaemorrhages, and the loss of normal capillary architecture [65,69]. Although most studies demonstrate improvement in capillaroscopic perfusion post-BTX-A treatment, its correlation with PROMs appears weak and inconsistent [13]. For example, Motegi et al. (*n* = 2 patients, systemic sclerosis) noted improvement in both Raynaud’s condition score and pain VAS as well as increased perfusion on CT angiography and nail fold capillaroscopy [32], while Du et al. (*n* = 16, systemic sclerosis) demonstrated a significant improvement in capillaroscopic findings in the BTX-A treatment group and with some improvement in the Quick-DASH, although this was not statistically significant [29]. Conversely, Shenavandeh et al. compared BTX-A to prostaglandin infusions in *n* = 16 and *n* = 10 patients, respectively, demonstrating improvements in clinical symptoms and a reduction in haemorrhages on capillaroscopy, but all other capillaroscopic features were unchanged [49]. The correlation between PROMs and objective outcome measures appears inconsistent across many trials, with some trials noting improvements in both measures and some showing discrepancies [6,13].

High-resolution Doppler ultrasound, applied to diffuse cutaneous SS patients versus healthy volunteers, identified decreased systolic velocities of digital arteries and nailfold microvasculature, potentially overlooked at more proximal levels or with lower Doppler frequencies [68]. Three studies assessed digital arterial perfusion or flow post-treatment with BTX-A. Blood flow velocity was demonstrated to increase from 30.5 (±14.4) cm/s before to 45.1 (±15.8) cm/s after treatment with BTX-A [48].

Laser Doppler imaging (LDI) is a non-invasive tool used to assess microvascular blood flow, previously employed in capturing response to temperature stimuli and vasodilators in patients with secondary Raynaud’s disease [64,69,71]. Based on the change in wavelength, as the laser exited tissue interacting with moving cells [64], it demonstrated a 58% increase in perfusion relative to untreated fingers in the original case report by Sycha et al. [10]. Subsequent studies, including by Neumeister et al., displayed varying responses in change in blood flow (−47 to 317%) after BTX-A [19]. In contrast, Bello et al., in the largest RCT assessing the clinical effectiveness of BTX-A in secondary Raynaud’s disease to date, revealed a reduction in perfusion by 30.08 flux units (7.7% from baseline) in patients allocated to the BTX-A group, with absolute blood flow at one-month lower than the placebo group (*p* = 0.018) [12]. While this study failed to meet clinical significance based on the MID, it is postulated that it included more treatment-resistant sub-groups of patients influencing outcomes [12]. These inconsistencies underscore the need to explore alternative modalities to sensitively capture changes in digital perfusion and flow post-BTX-A treatment.

#### 1.4.3. Other Objective Measures

Various objective measures of response to treatment have been reported in the literature. A frequently used measure is digital ulcer healing, with the TORCH study revealing all *n* = 18 studies demonstrated an improvement in digital ulcer healing, and six studies reported the complete resolution of digital ulceration [38]. Additionally, responses to the Cold Stimulation Test [2], skin temperature [42,46], oedema [40], range of motion as measured by the Kapandji thumb opposition test [37], and partial pressure of oxygen measured at the fingertip [6] have been explored.

## 2. Current Practice Guidelines

Despite the promising evidence supporting the clinical utility of BTX-A in Raynaud’s conditions of the hand, the lack of comprehensive clinical guidelines recommending its use poses a challenge for physicians striving to deliver standardised care. In clinical practice, the therapeutic benefit of BTX-A in refractory acute digital ulceration, the prevention of osteomyelitis, and digital preservation is recognised [6,13,38,72], although as noted by Geary et al., high-level evidence (such as from controlled trials) is limited [38].

While most plastic surgeons are familiar with BTX-A administration, including reconstitution and injection techniques, other clinicians may be more hesitant to prescribe it without clear clinical practice recommendations. A review of published British guidelines (BSSH/BHPR) reveals vague recommendations for the use of BTX-A in “severe/refractory ulcers in Raynaud’s phenomenon” in combination with sympathectomy [73]. The European guidelines (ESVM) do not address BTX-A in the treatment strategy for Raynaud’s conditions [74], while the American College of Rheumatology lists BTX-A as a 5th-line treatment in conjunction with sympathectomy [75]. The discrepancies between these international guidelines are likely to emanate from the lack of robust multicentre RCTs. Greater quantity and quality of evidence are needed to develop guidelines in this area. In addition to giving greater clarity for clinicians prescribing botulinum toxin, the recognition of botulinum toxin as a treatment for Raynaud’s phenomenon in guidelines may allow for organised training on the use of botulinum toxin for clinicians who would ordinarily not be familiar with it in their daily practice. While current coverage/reimbursement of botulinum toxin as a therapy for Raynaud’s phenomenon may vary between insurance providers/health services (potentially incurring high costs for patients), the recognition of this therapy in practice guidelines could reduce the financial burden on patients.

## 3. Future Directions

BTX-A may offer significant benefits to patients suffering from complications arising in Raynaud’s conditions of the hand, but there are key gaps in the literature that must be addressed to enable ease of adoption by clinicians. A consensus view on the use of BTX-A in Raynaud’s conditions of the hand, informed by international experts in rheumatology and hand surgery disciplines, may enable appropriately selected patients to benefit from this safe and effective locally acting drug. Delphi studies have previously been shown to help derive guidelines on new treatment strategies. The Delphi process can help minimise ‘groupthink’ to ensure a comprehensive assessment of the topic, particularly in the complex landscape of clinical decision-making [76]. In the absence of a clear consensus, a Delphi study provides a valuable framework for clinicians to make more informed decisions, clarifying areas of agreement and disagreement among experts [76]. This could help address the ambiguity often seen in the literature (for example, as noted above, the inconsistent response to BTX therapy when measured by PROMs or objective measures), as well as contribute insights that could enhance patient care in the pursuit of effective treatments for Raynaud’s conditions of the hand. There are no published Delphi studies on Raynaud’s conditions of the hand, although it has been used effectively in establishing guidelines in systemic sclerosis [77], indicating it may be applicable to this condition.

Integral to advancing the optimal use of BTX-A as a therapeutic avenue for Raynaud’s conditions of the hand is the establishment of a reliable and consistent method for outcome measurement. Indocyanine green (ICG)-enhanced fluorescence optical imaging (FOI) emerges as a potentially clinically useful tool for assessing perfusion in individuals with Raynaud’s conditions. This in vivo imaging modality visualises perfusion or inflammation by employing an injected fluorescence optical contrast media (ICG). Special light-emitting diodes excite the ICG dye, generating signals in the near-infrared spectrum that a specialised camera can detect. These real-time recorded images offer an assessment of perfusion [66,78,79]. Kang et al. demonstrated the utility of ICG-based imaging in diagnosing functional vascular insufficiency in the digits in as early as 2011 [66]. In 2017, Friedrich et al. employed ICG FOI to measure reduced perfusion to the digits affected by systemic sclerosis [80]. Before this, Pfiel et al. showcased in 2015 how FOI could objectively measure reduced inflammation post-iloprost and alprostadil administration [78]. Despite its utility in the assessment of connective tissue disease [78,79] and perfusion in other surgical settings [81,82,83,84], the use of FOI to objectively measure the effects of botulinum toxin in the treatment of Raynaud’s phenomenon has yet to be explored.

Finally, refining our understanding of the basic mechanism of action of BTX-A in Raynaud’s conditions of the hand will be crucial for clinicians attempting to deliver a consistent and reliable treatment response. Future studies of BTX-A in vitro, in explant tissue, and in preclinical models will help optimise formulation, dosage, concentration, mode of delivery and frequency of administration for patients requiring treatment. One example of this has been information from previous animal studies to give insight into the effects of BTX-A on wounds and blood flow [20,25,26,28].

## 4. Conclusions

The treatment of Raynaud’s conditions, both with and without systemic autoimmune disease, is extremely challenging. Currently, most pharmacological and surgical strategies are hindered by inefficacy, inconsistency, and high adverse side effect profiles, making optimal treatment strategies for this condition an important area of unmet clinical need. [1,2,3,5,7,15]. BTX-A treatment appears to avoid systemic side effects, is well tolerated, and appears clinically effective, although robust high-level evidence of this is limited. Widespread adoption of this therapeutic strategy and its inclusion in clinical guidelines must be informed by high-quality clinical trial data.

To accurately assess the clinical effectiveness of BTX-A in future robust studies, patient selection must be streamlined to minimise heterogeneity in disease severity, recruitment must be maximised to ensure adequate statistical power, and optimal outcome measures must be selected that are sensitive enough to monitor subtle changes in treatment response to discern treatment efficacy. Additionally, addressing concerns such as minimising antibody resistance, drug development to extend the duration of treatment effects on tissues, and exploring new modalities to accurately assess hand perfusion should be focal points for future research.

## Figures and Tables

**Figure 1 toxins-16-00472-f001:**
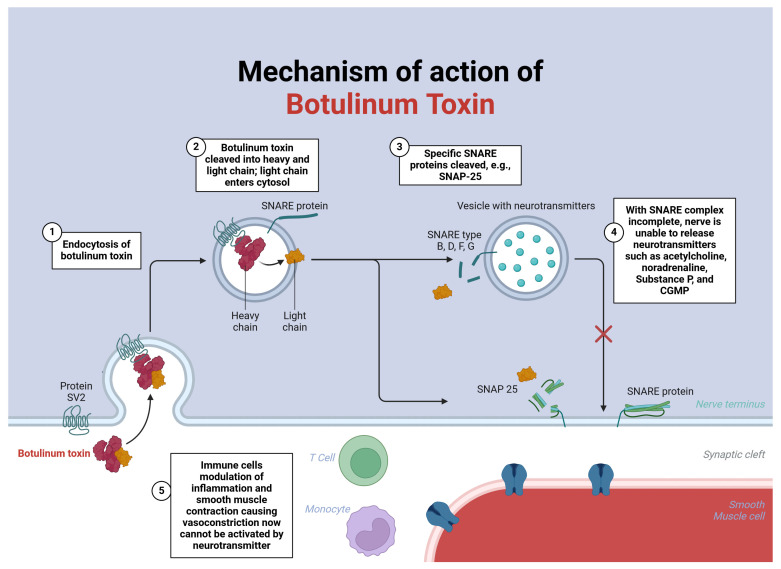
Mechanism of action of Botulinum Toxin.

**Figure 2 toxins-16-00472-f002:**
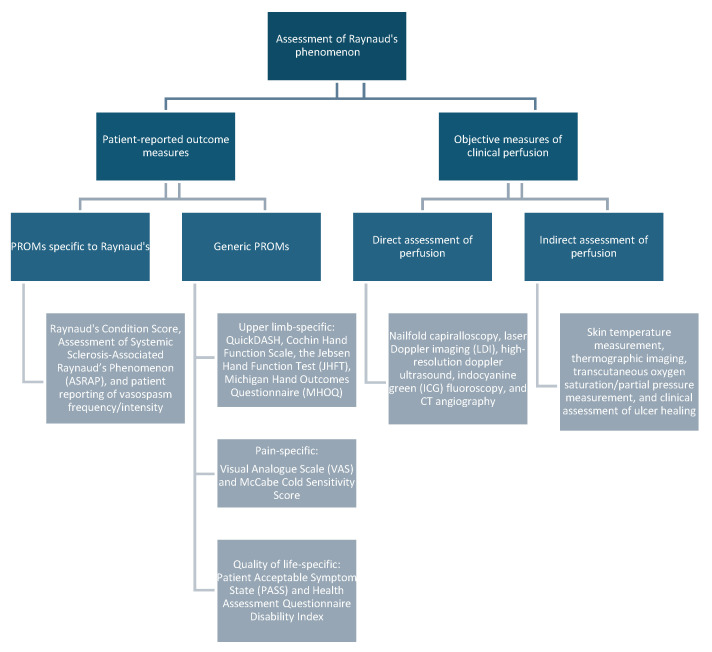
Outcome measures for assessment of Raynaud’s phenomenon.

## Data Availability

The data presented in this study are available in this article.

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
