# Peer review of "Botulinum Toxin for the Treatment of Raynaud’s Conditions of the Hand: Clinical Practice Updates and Future Directions"

_toxins, 2024, doi:10.3390/toxins16110472_

Round 1

Reviewer 1 Report

Comments and Suggestions for Authors

The manuscript is generally well-written, but requires enrichment in a few specific areas:

1.     Please create a figure/diagram that illustrates the proposed mechanisms of action. The paper feels arid at this point. 

2.     The current mechanistic discussion of BTX-A’s role in Raynaud’s is somewhat limited for a paper that aims to explore therapeutic pathways in depth. This section needs to be expanded with more data from the literature.

For instance, in psoriasis, it has been shown that BTX-A also reduces the release of neuropeptides such as Substance P and CGRP, both of which play roles in neurogenic inflammation. So BTX-A’s modulation of neuroimmune activity has shown efficacy in reducing pain and inflammation. And this could also be useful in Raynaud’s.

Please see this paper: Popescu, M.N.; Beiu, C.; Iliescu, M.G.; Mihai, M.M.; Popa, L.G.; Stănescu, A.M.A.; Berteanu, M. Botulinum Toxin Use for Modulating Neuroimmune Cutaneous Activity in Psoriasis. Medicina 2022, 58, 813. https://doi.org/10.3390/medicina58060813 

3.     Also, please include a section discussing the disadvantages and practical challenges associated with BTX-A therapy in Raynaud’s phenomenon, such as high costs (BTX-A treatments are often not reimbursed by insurance, making them expensive for patients), pain during injection (administering multiple injections into sensitive areas like the distal palm and fingers can be quite painful for patients), or lack of specialized training (physicians are not typically trained in BTX-A administration during residency, requiring additional personal training for proficiency)

Author Response

Comments and Suggestions for Authors

The manuscript is generally well-written, but requires enrichment in a few specific areas:

  1. Please create a figure/diagram that illustrates the proposed mechanisms of action. The paper feels arid at this point. 

 Many thanks for this suggestion. The authors would be happy to create such a diagram, however with only a short time period to respond to these initial comments, we have not been able to get assistance in graphic design to produce the best quality figure possible. The editor of Toxins has stated that they are willing to give the authors a few more days to prepare a figure to illustrate this point and we will update with this figure as soon as it is ready

  1. The current mechanistic discussion of BTX-A’s role in Raynaud’s is somewhat limited for a paper that aims to explore therapeutic pathways in depth. This section needs to be expanded with more data from the literature.

For instance, in psoriasis, it has been shown that BTX-A also reduces the release of neuropeptides such as Substance P and CGRP, both of which play roles in neurogenic inflammation. So BTX-A’s modulation of neuroimmune activity has shown efficacy in reducing pain and inflammation. And this could also be useful in Raynaud’s.

Please see this paper: Popescu, M.N.; Beiu, C.; Iliescu, M.G.; Mihai, M.M.; Popa, L.G.; Stănescu, A.M.A.; Berteanu, M. Botulinum Toxin Use for Modulating Neuroimmune Cutaneous Activity in Psoriasis. Medicina 2022, 58, 813. https://doi.org/10.3390/medicina58060813 

Many thanks for this suggestion. Our previous discussion on the mechanism of action (lines 113-130 in the original manuscript) had made mention of Substance P briefly but we note this paper with interest and have added a line citing it and its data. Otherwise we have expanded the language in these paragraphs to address these concerns (lines 111-135 in the edited manuscript). The authors intended focus for this paper was for a broad overview of the application of botulinum toxin as a therapeutic option in Raynaud’s disease with a particular focus on assessing outcomes through currently available methods, hence the shorter section on specific mechanisms of action.

  1. Also, pleaseinclude a section discussing the disadvantages and practical challenges associated with BTX-A therapy in Raynaud’s phenomenon, such as high costs (BTX-A treatments are often not reimbursed by insurance, making them expensive for patients), pain during injection (administering multiple injections into sensitive areas like the distal palm and fingers can be quite painful for patients), or lack of specialized training (physicians are not typically trained in BTX-A administration during residency, requiring additional personal training for proficiency)

Many thanks for highlighting this. Safety concerns regarding pain are specifically discussed in the original manuscript in lines 192-200, however we have expanded the language here to emphasise this point. In lines 321-323 of the original manuscript the authors had noted issues with clinicians other than plastic surgeons being familiar with botulinum toxin, but we have added language to this section to add training (lines 346-350 in the edited manuscript). We have also added a point regarding cost at this point (lines 350-353 in the edited manuscript).

Reviewer 2 Report

Comments and Suggestions for Authors

See attached.

Author Response

This is a clearly written narrative review of the use of botulinum toxin in people with Raynaud’s condition of the hands.

It is generally a comprehensive review of the topic and appropriate recommendations for future research design are made.

However, more could be said about the weaknesses of the current research for this clinical condition generally. The overall evidence for efficacy of botulinum toxin in people with Raynaud’s condition of the hands is marginal (except possibly for healing of digital ulcers), or not established, in controlled trials.

The systematic review that has been referenced (the TORCH study) had as its major recommendation that an adequately powered randomised trial is conducted.

This review should be more explicit about the limited evidence of efficacy in this clinical condition and clearer about the need for appropriately designed and powered clinical trials. Those trials should also carefully record adverse effects of treatment.

Line 102 – other botulinum toxin preparations are not mentioned.

Many thanks to the reviewer for their comments. We have added language to emphasise the weakness of the current literature at lines 188-190 in the edited manuscript and again at lines 335-336, 344-346, and 401. The authors aim for this paper was to focus on outcome measures so as to explore how best to design future trials in this area.

Regarding the second point we have added to the sentence at line 104-105 in the edited manuscript to include incobotulinumtoxinA (incoBoNT-A) = Xeomin®, and rimabotulinumtoxinB = Myobloc®. If there are other preparations the reviewer feels it would be appropriate to mention please advise on these.

Reviewer 3 Report

Comments and Suggestions for Authors

General considerations:

This paper reviewed the literature on the efficacy and dosage of botulinum toxin in diseases presenting Reynaud phenomenen.

The topic is interesting, considering the scarce evidences of this indication that has an impact on quality of life of patients.

However, I have several concerns:

1.                  Paragraph 1.1: In my opinion the paragraph is overly verbose and this part should be deleted “The primary BTX-A products utilised in clinical practice are onabotulinumtoxinA (marketed as Botox®) and abobotulinumtoxinA (marketed as Dysport®). Both are derived from different strains of Clostridium botulinum and function by impeding the transmission of motor and autonomic nerve signals through the inhibition of presynaptic acetylcholine release. Notably only BTX-A and B are approved for human use- BTX-A targets SNAP-25 and BTX-B targets VAMP. Botox® received FDA approval in 2002, preceding Dysport® by 7 years. Over the last 4 decades, Botox® has been employed as a therapeu-109tic agent in various conditions, including autonomic disorders, spasticity, migraine and 110neuropathic pain, cosmetic treatment of rhytides, hyperhidrosis, overactive bladder, and Raynaud’s conditions.”

2.               I believe that a schema could help to recognize the subjective (patient-reported outcome measures) and objective measures of the disease.  

Author Response

General considerations:

This paper reviewed the literature on the efficacy and dosage of botulinum toxin in diseases presenting Reynaud phenomenen.

The topic is interesting, considering the scarce evidences of this indication that has an impact on quality of life of patients.

However, I have several concerns:

  1. Paragraph 1.1: In my opinion the paragraph is overly verbose and this part should be deleted “The primary BTX-A products utilised in clinical practice are onabotulinumtoxinA (marketed as Botox®) and abobotulinumtoxinA (marketed as Dysport®). Both are derived from different strains of Clostridium botulinum and function by impeding the transmission of motor and autonomic nerve signals through the inhibition of presynaptic acetylcholine release. Notably only BTX-A and B are approved for human use- BTX-A targets SNAP-25 and BTX-B targets VAMP. Botox® received FDA approval in 2002, preceding Dysport® by 7 years. Over the last 4 decades, Botox® has been employed as a therapeu-109tic agent in various conditions, including autonomic disorders, spasticity, migraine and 110neuropathic pain, cosmetic treatment of rhytides, hyperhidrosis, overactive bladder, and Raynaud’s conditions.”

Many thanks for this feedback. The above paragraph has been reduced, although we have left a shorter edited version in order to address concerns from another reviewer (see lines 102-110 in the edited manuscript).

  1.               I believe that a schema could help to recognize the subjective (patient-reported outcome measures) and objective measures of the disease.

 Many thanks for this suggestion. A sample schema has been added. This can be found in the edited manuscript at line 241.